# Physiological Characteristics of Root Regeneration in Rice Seedlings

**Yizhuo Gao** [1,2,†], **Yan Zhu** [3,†], **Yuping Zhang** [1], **Yikai Zhang** [1], **Yaliang Wang** [1], **Zhigang Wang** [1], **Huizhe Chen** [1,*], **Yunbo Zhang** [2,*] and **Jing Xiang** [1,*]

1   State Key Laboratory of Rice Biology, China National Rice Research Institute, Hangzhou 310006, China; 15353871810@163.com (Y.G.); cnrrizyp@163.com (Y.Z.); yikaizhang168@163.com (Y.Z.); wangyaliang@caas.cn (Y.W.); zgwang212@126.com (Z.W.)
2   MARA Key Laboratory of Sustainable Crop Production in the Middle Reaches of the Yangtze River, College of Agriculture, Yangtze University, Jingzhou 434025, China
3   Linping District Agricultural Technique Extension Station, Hangzhou 311100, China; zhuyan221@126.com
*   Correspondence: chenhuizhe@163.com (H.C.); yzhang@yangtzeu.edu.cn (Y.Z.); xiangjing@caas.cn (J.X.); Tel.: +86-153-5546-0231 (H.C.); +86-150-2704-4560 (Y.Z.); +86-189-5807-1661 (J.X.)
†   These authors contributed equally to this work.

**Abstract:** The rapid development of new roots in transplanted rice (*Oryza sativa* L.) is crucial for shortening the returning green time of seedlings and accelerating tillering. Root regeneration plays an important role in enabling seedlings to resume normal growth and produce effective spikes after root injury. This study aimed to investigate the dynamic changes in new root production and the growth of seedlings of different varieties after root cutting in addition to the key physiological factors. We utilized hydroponics to set up four different time treatments to observe the occurrence of root systems in various rice seedling varieties after root cutting; we also measured related physiological indexes to further analyze the results. This study found that changes in aboveground nutrient, energy, and hormone levels in seedlings are critical for the growth of new roots after cutting. A morphological analysis showed that the root germination force of Zhongzao 39 (ZZ39) was stronger than Jiazao 311 (JZ311) before shearing and weaker after shearing. Physiological and biochemical analyses revealed that both ZZ39 and JZ311 experienced a decrease in their aboveground nitrogen and phosphorus content after root cutting. Soluble sugar content and starch content were found to decrease to their lowest levels after two days of root shearing. Furthermore, both varieties showed a significant increase in aboveground indole-3-acetic acid (IAA) content after two days of root shearing, and the IAA content in new roots was also higher. The results indicate that higher levels of hormones in seedlings with cut roots can enhance the transportation of nutrients and carbohydrates from the stems and leaves to the roots, leading to improved growth and the production of new roots. Additionally, the accumulation of IAA in damaged roots can also positively impact this process. This study found significant differences in the regeneration of rice seedling roots after cutting depending on the variety. We identified key physiological characteristics that affect new root generation, which provides a scientific basis for identifying strong root regeneration varieties and developing cultivation measures to promote new root growth in rice.

**Keywords:** rice seedling; hydroponics; root regeneration; physiological characteristics

## 1. Introduction

Rice (*Oryza sativa* L.) is the predominant food crop in China, with approximately 60% of the population relying on it for sustenance [1]. As social and economic development has progressed and rural labor has migrated, traditional manual rice transplanting methods have become outdated and unsuited for modern rice crop technology. Additionally, advancements in agricultural machinery and agronomic technology have led to the rapid development of rice seedling machine transplanting as the primary planting method [2–4].

The current rice machine transplanting technology in China involves centralized seedling breeding in seedling trays and inserting seedlings through seedling claws. However, this process results in extensive damage to the root system of rice seedlings. Rice roots play a vital role in the growth and development of the plant by synthesizing, sequestering, absorbing, and storing essential nutrients [5]. Furthermore, the roots' morphological and physiological characteristics have a significant impact on the yield and quality of rice [6–10].

According to research, machine transplanting may hinder the early development and fast growth of seedlings due to prolonged delays in the seedling period and reproductive process when compared with traditional manual transplanting [2]. Root regeneration force refers to the ability of plants to produce new roots and resume growth after root damage.

In China and abroad, the length or weight of new roots in a single plant is commonly used as an index of root regeneration force [11,12]. Research has shown that by enhancing the root regeneration force of rice, the seedling returning green time can be shortened and the early growth and rapid development of tillers can be promoted [13]. Root regeneration is a crucial factor in assessing the quality of seedlings, as highlighted by numerous studies [14–17].

The production of new roots in seedlings is influenced by exogenous substances, as well as factors such as the variety, cultivation environment, seedling nursery, and planting method [11,14,18]. In a study conducted by Takahashi et al. [19], it was demonstrated that the application of moderate amounts of sucrose, glucose, and fructose artificially induced the production of adventitious roots in plants; however, the application of mannose or sorbitol did not have the same effect. Other studies have also suggested that the presence of nitrate nitrogen in the growth medium can stimulate the production of lateral roots and promote an increase in root length [20,21]. A previous study indicated that the primary hormone responsible for initiating the formation of new roots is auxin [22]. According to Correa et al. [23], exogenous indole-3-acetic acid (IAA) was found to have a positive effect on the rooting of excised leaves and etiolated seedlings.

While numerous studies have explored the impact of external factors on plant root initiation, there is a dearth of research on the intrinsic physiological characteristics of machine-planted wounded seedlings and their mechanisms for new root germination [24,25]. This study aimed to investigate the new root growth dynamics and physiological changes in rice seedlings after root injury using two experimental materials, ZZ39 (the weak-root regeneration variety) and JZ311 (strong-root regeneration variety). These results will provide a scientific and theoretical basis for exploring the key factors affecting new root production and will help formulate cultivation measures to improve the root regeneration of machine-planted seedlings in different varieties.

## 2. Materials and Methods

### 2.1. Plant Materials and Growth Conditions

In April 2022, a study was conducted at China's National Rice Research Institute to grow rice seedlings using the Yoshida total nutrient solution. The seedlings were grown in a plant growth chamber with controlled temperature, humidity, and light intensity. After screening over 20 materials, we identified JZ311 as a strong-root regeneration variety and ZZ39 as a weak-root regeneration variety through multiple rounds of screening. The incubator temperature was maintained between 22 and 25 °C with a humidity level of 70%. The light intensity was adjusted based on different time periods, with natural light intensity ranging from 06:00 to 19:00 and dark treatment ranging from 19:00 to 00:00 and from 00:00 to 06:00.

### 2.2. Experimental Design

Rice seeds with consistent germination were planted in 96 (12 × 8)-well plates, with each well having a diameter of 3 mm. In each time gradient, 5 plates were sown, with a total of 48 seeds being sown per plate. The 96-well plates were black and were placed in a 10.5 cm high black opaque hydroponic box. Nutrient solution was added to the hydroponic box until the seeds were just submerged. The nutrient solution was prepared using the

Yoshida total nutrient solution formula, which was changed every 3–4 days. The pH of the solution was adjusted between 5.3 and 5.5 using dilute hydrochloric acid.

The experiment consisted of four time treatment gradients, which were 0, 2, 4, and 6 d. The day 0 treatment involved the removal of all roots and the collection of the seedlings when they reached the three-leaved stage. For the day 2 treatment, all new roots were cut off and collected after 2 d of hydroponics. Similarly, for the day 4 and 6 treatments, all new roots were cut off and collected after 4 and 6 d of hydroponics, respectively. The experiment included a total of 8 treatments, each with 3 replications, and lasted for 20 days.

### 2.3. Measurement Items and Methods

For each period, we collected new roots from both varieties and scanned them using an Epson Perfection V700 Photo plant root scanner. The resulting images were stored in a computer and analyzed using the Win-RHIZO PRO 2013 root analysis system software to obtain parameters such as total root length, root number, root surface area, and root volume for each treatment. In each period, we selected thirty seedlings of each variety with uniform growth, and each ten seedling set was used as one replicate. We then separated the aboveground and root systems, dried them to constant weight, and weighed their dry matter mass. In order to measure the mass fraction of total nitrogen, total phosphorus, and total potassium, the seedling stems and leaves were dried and crushed. Three replicates were set up, and 0.1 g of each replicate was weighed as the sample. An automatic Kjeldahl nitrogen tester (Foss) was used for the measurement of total nitrogen [26].

The molybdenum-antimony anti-colorimetric method was used for the measurement of total phosphorus, and the flame photometric method was used for the measurement of total potassium. For the determination of soluble sugars and starch, the anthrone colorimetric method was used [27]. In this study, we extracted chlorophyll and carotenoids from 0.1 g of fresh rice seedlings using 96% ethanol. The absorbance of the extract was measured at 665, 649, and 470 nm using a spectrophotometer, following the method described by Lichtenthaler and Wellburn [28]. For each treatment, ten seedlings of uniform growth were selected, and the plants and roots were separated. High-performance liquid chromatography (HPLC) [29] at each treatment period determined the GA3, IAA, ABA, and ZR content of seedlings in the aboveground and roots portions.

### 2.4. Statistical Analysis

In this study, the data were organized and calculated using Excel 2021 software, and SPSS 23.0 statistical software was used to perform three repeated one-way analyses of variance (ANOVA) and Duncan's multiple comparisons tests. All values were reported as the means $\pm$ SD (standard deviation). Pictures were drawn using SigmaPlot 10.0 software.

## 3. Results

### 3.1. Changes in Seedling Quality at Different Periods

The aboveground and root dry matter mass increased with time after root cutting in both varieties (Figure 1). Before root pruning, there was no significant difference in aboveground dry matter mass between ZZ39 and JZ311. However, subsequent to root pruning, JZ311 demonstrated a considerably greater aboveground dry matter mass compared with ZZ39 (Figure 1A). Initially, ZZ39 had a higher root dry weight compared with JZ311. After root pruning, ZZ39's root dry weight significantly decreased and became smaller than JZ311's root dry weight. On the sixth day, there was no significant difference in the dry weight of new roots between the two varieties (Figure 1B). In addition, the aboveground dry matter mass of ZZ39 exhibited the highest growth rate of 6 days after root cutting, with an increase of 50.34% compared with day 4. On the other hand, JZ311 showed the highest growth rate on day 4, with an increase of 49.61% compared with 2 days after cutting (Figure 1A). On day 4, ZZ39 and JZ311 exhibited the highest rate of growth in new root dry weight with values of 10.65 and 13.37 mg/d, respectively (Figure 1B).

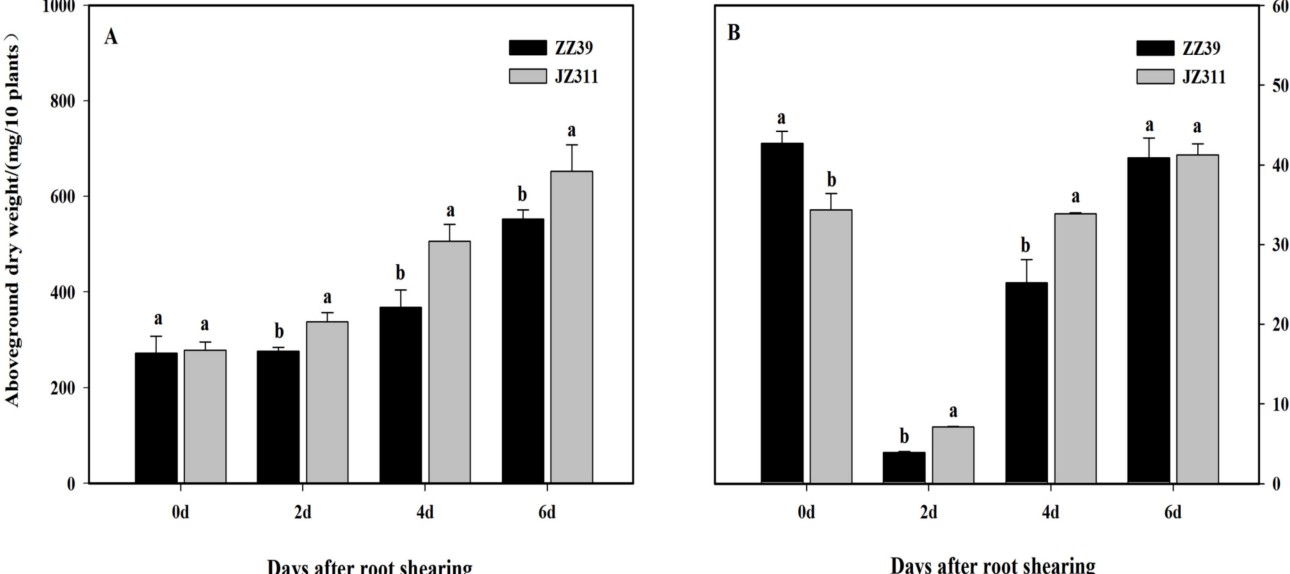

**Figure 1.** Aboveground and root dry weight of rice seedlings at different treatment periods: (**A**) the aboveground dry weight of ZZ39 and JZ311 on days 0–6; (**B**) the root dry weight of ZZ39 and JZ311 on days 0–6. The columns with the same letter are not significantly different at $p < 0.05$.

The trends in root length, root tip number, root surface area, and volume over time were similar for both ZZ39 and JZ311 (Figure 2). After root pruning, the growth rate of a single plant's new root length was found to be the fastest in the 2–4 day period for both ZZ39 and JZ311 at 12.4 cm/d (Figure 2A,E). In addition, the number of single root tips for both varieties significantly increased at a rate of 14.5 and 14.45 tips/d, respectively (Figure 2B). The root surface area for ZZ39 and JZ311 increased the most 4 days after cutting, measuring 0.36 and 0.425 $cm^2$/d, respectively (Figure 2C,E). Furthermore, the new root volume also significantly increased during this period, with measurements of 0.00475 and 0.0072 $cm^3$/d for ZZ39 and JZ311, respectively (Figure 2D).

*3.2. Changes in Aboveground Soluble Sugar and Starch Content of Rice Seedlings at Various Periods*

The soluble sugar content and starch content in the aboveground ZZ39 and JZ311 seedlings from days 0 to 6 showed similar trends (Figure 3). At days 0 and 2, the aboveground soluble sugar content of ZZ39 seedlings was significantly higher than that of JZ311, but at days 4 and 6, it was significantly lower than that of JZ311 (Figure 3A). On the other hand, the aboveground starch content of JZ311 seedlings was not significantly different from ZZ39 at days 0 and 2, but it was significantly higher than ZZ39 at days 4 and 6 (Figure 3B).

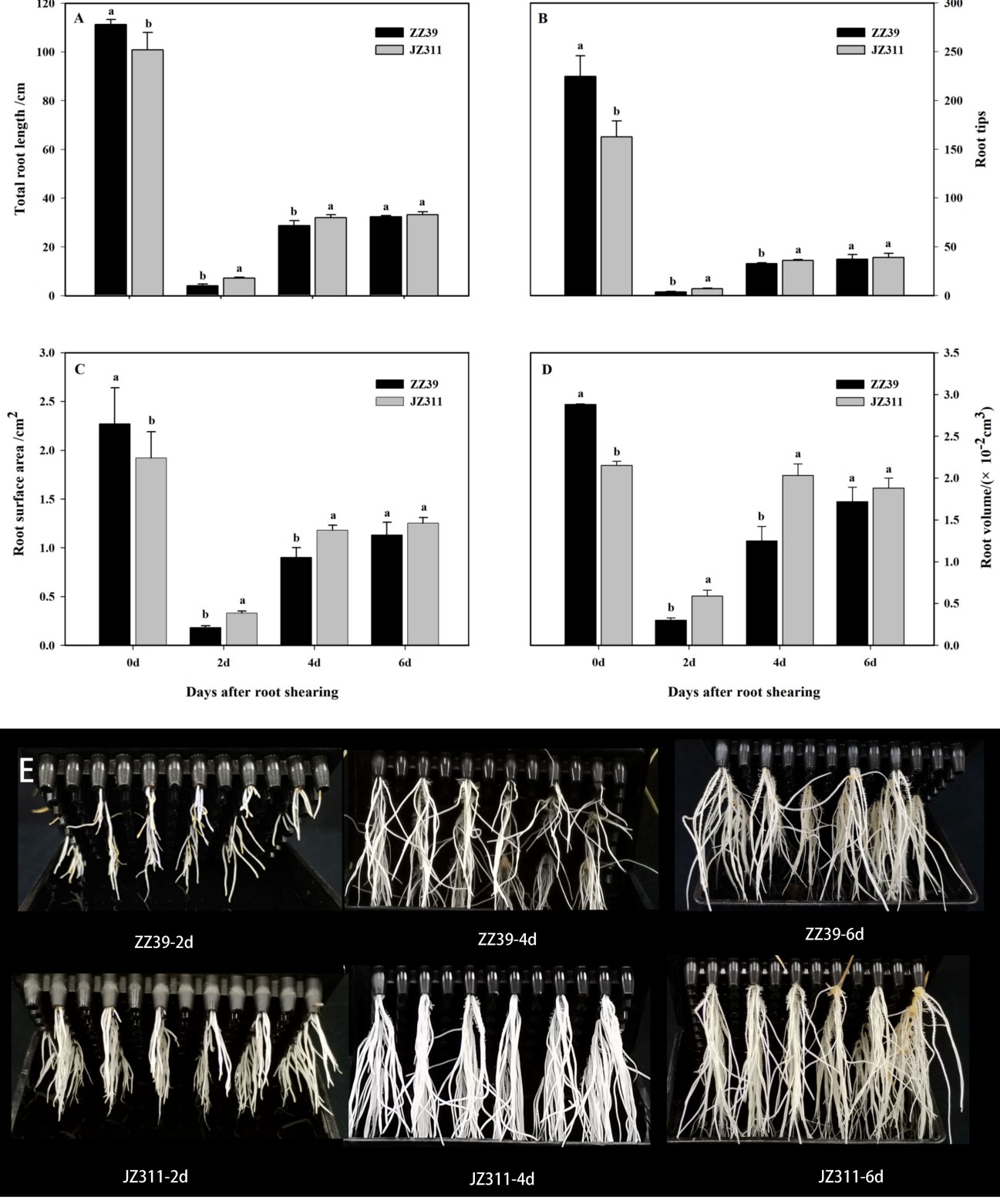

**Figure 2.** Root traits of rice seedlings at different treatment periods: (**A**) the total root length of a single plant of ZZ39 and JZ311 on days 0–6; (**B**) the root tips of a single plant of ZZ39 and JZ311 on days 0–6; (**C**) the root surface area of a single plant of ZZ39 and JZ311 on days 0–6; (**D**) the root volume of a single plant of ZZ39 and JZ311 on days 0–6. The columns with the same letter are not significantly different at $p < 0.05$. (**E**) The three columns show the changes in new roots 2, 4, and 6 d after cut-root hydroponics. The two rows represent the ZZ39 and JZ311 varieties.

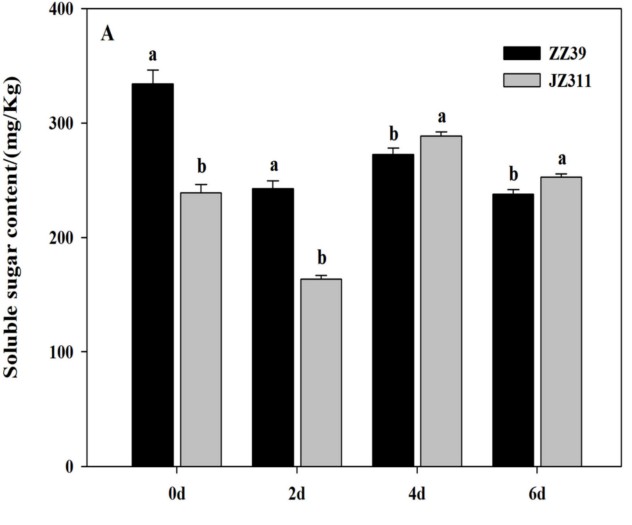
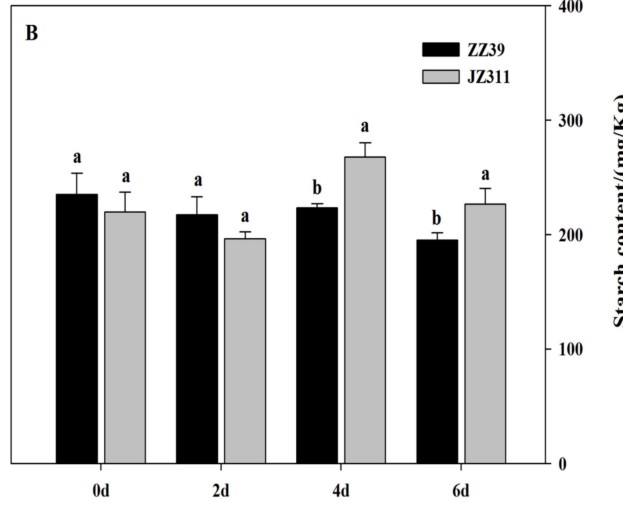

**Figure 3.** Soluble sugar and starch content of rice seedlings at different treatment periods: (**A**) the soluble sugar content of ZZ39 and JZ311 on days 0–6; (**B**) the starch content of ZZ39 and JZ311 on days 0–6. The columns with the same letter are not significantly different at *p* < 0.05.

The soluble sugar content of ZZ39 and JZ311 seedlings significantly decreased by 27.4% and 31.6%, respectively, after 2 days of root cutting compared with the initial day; however, the soluble sugar content rapidly increased after 4 days of root cutting by 12.2% and 76.4%, respectively, compared with the levels after 2 days. The content then decreased on day 6 compared with day 4 by 12.7% and 12.5%, respectively (Figure 3A). After 2 days of root pruning, the aboveground starch content of seedlings from ZZ39 and JZ311 decreased by 7.5% and 10.6%, respectively, compared with the initial measurement. Nevertheless, after 4 days of root cutting, JZ311 showed a substantial increase of 36.1% compared with the measurement taken 2 days after cutting, whereas ZZ39 only exhibited an increase of 2.6% (Figure 3B).

### 3.3. Changes in the Aboveground Nutrient Content of Rice Seedlings at Various Periods

The aboveground nitrogen content of ZZ39 and JZ311 seedlings showed significant variations across different treatment periods (Figure 4A). Before root pruning, ZZ39 and JZ311 had the highest nitrogen content, but this declined to its lowest level after 4 days of root cutting. The nitrogen content in the aboveground portion of ZZ39 and JZ311 seedlings decreased the most after 4 days, with reductions of 16.5% and 30.8%, respectively, compared with the initial measurement. However, after 6 days of root shearing, the aboveground nitrogen content in both varieties increased by 11.4% and 20.9%, respectively, compared with the measurement taken 4 days after cutting.

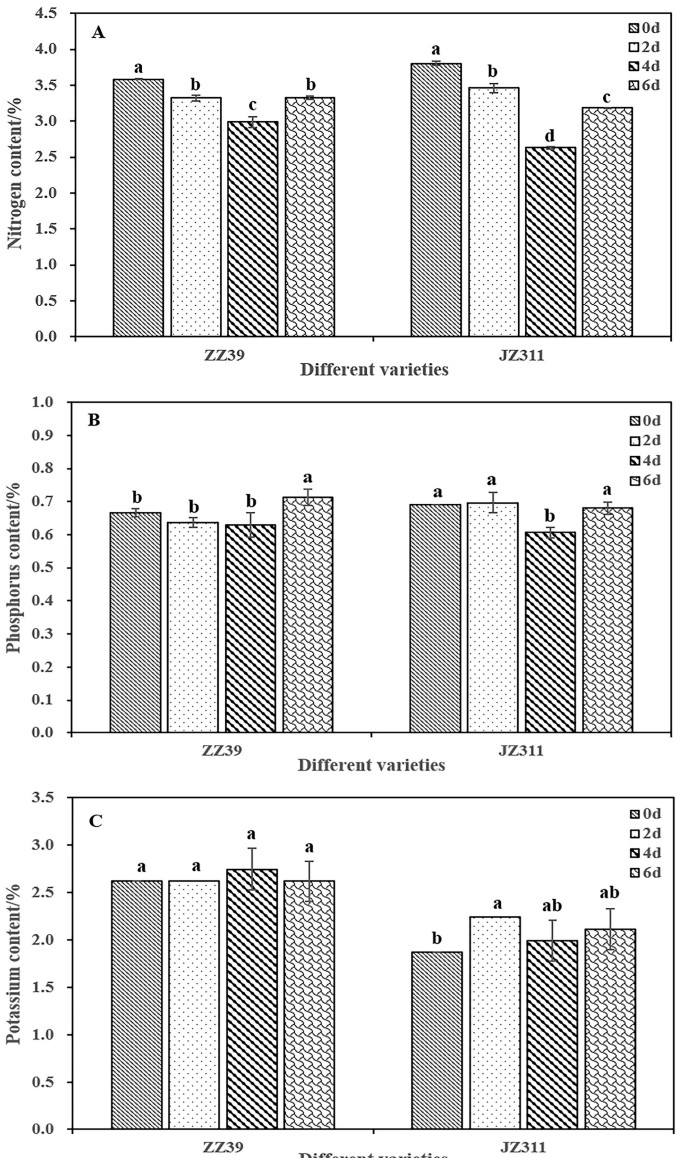

**Figure 4.** Nutrient content of rice seedlings at different treatment periods: (**A**) the nitrogen content of ZZ39 and JZ311 on days 0–6; (**B**) the phosphorus content of ZZ39 and JZ311 on days 0–6; (**C**) the potassium content of ZZ39 and JZ311 on days 0–6. The columns with the same letter are not significantly different at $p < 0.05$.

Although there was only a slight difference in the aboveground phosphorus content of ZZ39 seedlings between day 0 and day 4, it was significantly lower compared with day 6. Similarly, the aboveground phosphorus content of JZ311 seedlings was significantly lower 4 d after cutting than at all other periods (Figure 4B). The aboveground phosphorus content of ZZ39 seedlings gradually decreased by 4.5% and 5.5% during the period of days 0–4 compared with day 0; however, it showed a rapid recovery by increasing by 7% 6 d after cutting. On the other hand, the phosphorus content aboveground in JZ311 seedlings remained stable during the initial 0–2-day period but significantly decreased by 12.1% on day 4 after cutting compared with day 0.

The aboveground potassium content of ZZ39 seedlings showed no significant difference among various treatment periods, although it was the highest 4 d after shearing. In contrast, the aboveground potassium content of JZ311 seedlings was significantly higher 2 d after shearing, having a 19.8% increase compared with day 0 (Figure 4C).

### 3.4. Changes in the Aboveground Pigment Content of Rice Seedlings at Various Periods

The study found that JZ311 seedlings had significantly higher levels of chlorophyll-a,-b, carotenoids, and total pigments compared with ZZ39 before root pruning. However, six days after cutting, the difference was not significant (Figure 5). The aboveground chlorophyll-a content of ZZ39 seedlings increased by 61.3% two days after shearing but decreased by 38.8% and 56.0% at four and six days after shearing, respectively. The chlorophyll-a content of JZ311 seedlings was highest before root shearing but then significantly decreased by 10.4%, 30.5%, and 63.6% at 2, 4, and 6 days after shearing, respectively, compared with the measurement on day 0 (Figure 5A). There was no significant difference in the chlorophyll-b content of ZZ39 seedlings between day 0 and 2 after shearing. However, it significantly decreased by 32.0% and 51.4% 4 d and 6 d after shearing, respectively, when compared with 2 d after shearing (Figure 5B).

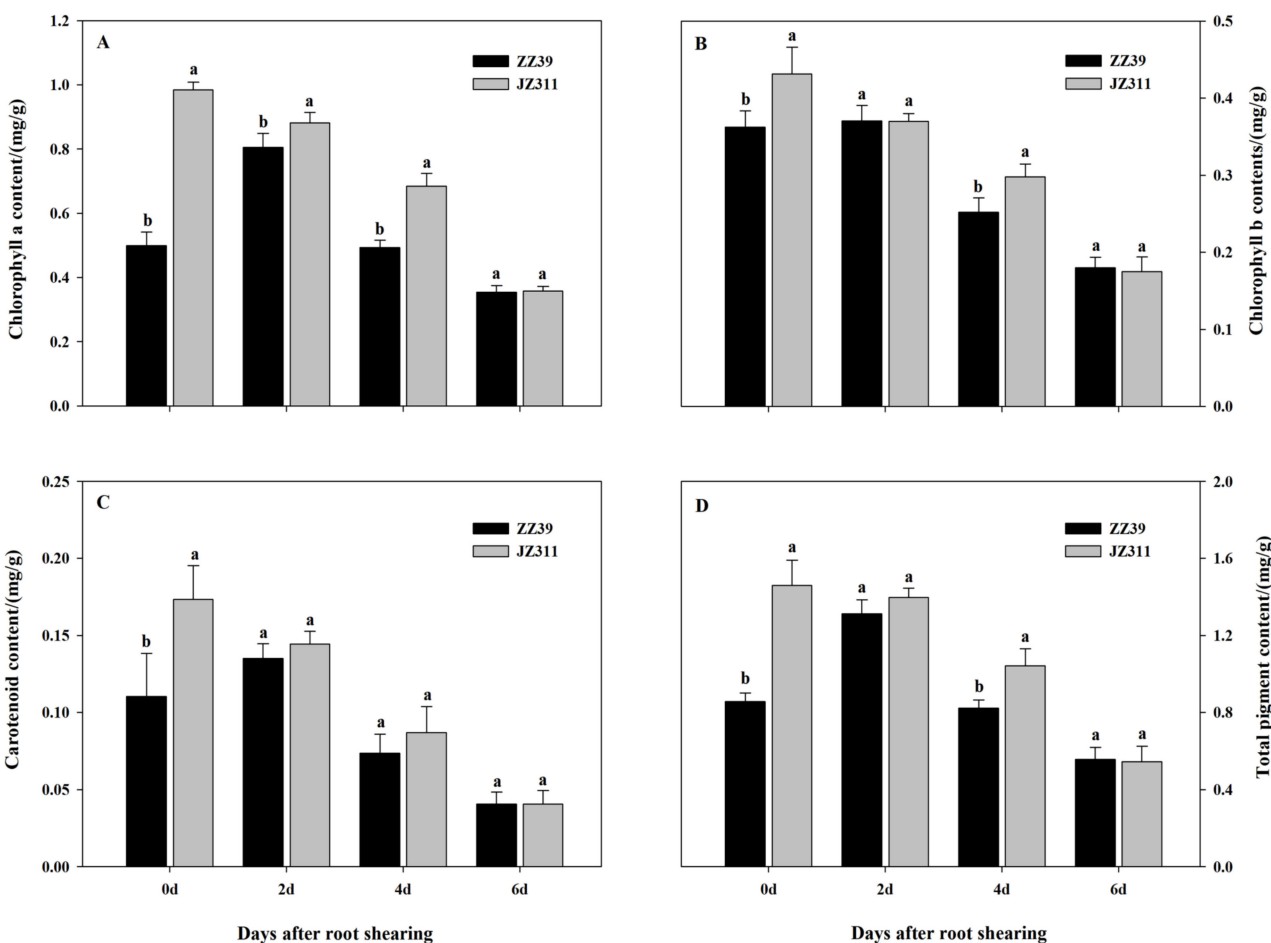

**Figure 5.** Different pigmentation levels of rice seedlings at various treatment periods: (**A**) the chlorophyll-a content of ZZ39 and JZ311 on days 0–6; (**B**) the chlorophyll-b content of ZZ39 and JZ311 on days 0–6; (**C**) the carotenoid content of ZZ39 and JZ311 on days 0–6; (**D**) the total pigment content of ZZ39 and JZ311 on days 0–6. The columns with the same letter are not significantly different at $p < 0.05$.

The carotenoid content of ZZ39 seedlings increased by 22.2% 2 days after shearing compared with before the cutting processing, but then decreased by 45.5% and 70.0% 4 and 6 days after shearing, respectively. The carotenoid content of JZ311 seedlings decreased by 16.7%, 49.8%, and 76.6% on days 2, 4, and 6, respectively, after shearing compared with the initial level (Figure 5C). The total pigment content of ZZ39 seedlings significantly increased after 2 days, but then experienced decreases of 37.3% and 57.5%. Meanwhile, the total pigment content of JZ311 seedlings experienced decreases of 4.4%, 28.7%, and 62.7% after

0 days (Figure 5D). This trend indicates a significant reduction in the plant's ability to synthesize pigments.

### 3.5. Changes in Aboveground Hormone Content of Rice Seedlings at Different Periods

The aboveground content of GA3, IAA, and ABA was not significantly different in the ZZ39 and JZ311 seedlings before root shearing. Four days after root cutting, the aboveground content of GA3, ABA, and ZR in the ZZ39 seedlings was significantly higher than those in the JZ311 seedlings. After 6 days, there was no significant difference in the levels of aboveground IAA and ABA in the seedlings of the two varieties; however, the levels of GA3 and ZR were found to be significantly different (Figure 6).

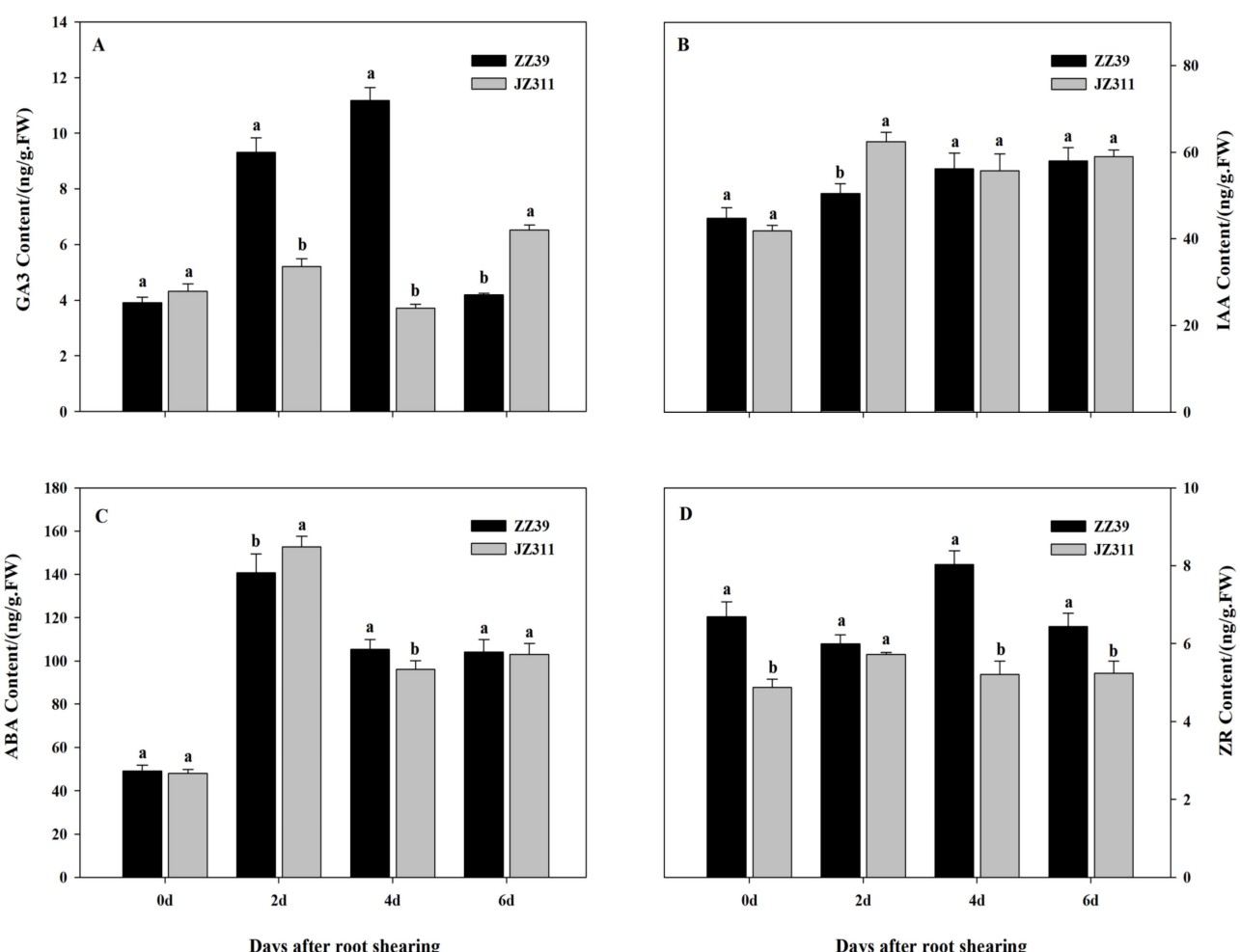

**Figure 6.** Different hormone content in the aboveground parts of rice seedlings at various periods: (**A**) the GA3 content of ZZ39 and JZ311 on days 0–6; (**B**) the IAA content of ZZ39 and JZ311 on days 0–6; (**C**) the ABA content of ZZ39 and JZ311 on days 0–6; (**D**) the ZR content of ZZ39 and JZ311 on days 0–6. The columns with the same letter are not significantly different at $p < 0.05$.

The study found that the GA3 content in ZZ39 seedlings increased by 1.38 times on day 2 and 1.86 times on day 4 after root cutting compared with day 0. However, the GA3 content sharply decreased after 6 days, showing a 62.5% decrease compared with day 4. The aboveground GA3 content of JZ311 seedlings showed a 20.5% increase on day 2 compared with Day 0, but significantly decreased by day 4 (Figure 6A).

The concentration of IAA in the aboveground of ZZ39 seedlings increased by 12.7%, 25.4%, and 29.5% after 2, 4, and 6 days, respectively, compared with the initial concentration of day 0. The levels of aboveground IAA in JZ311 seedlings showed a significant increase

of 49.3% after 2 days compared with 0 days; however, there was a subsequent decrease of 10.7% at 4 days compared with the levels observed on day 2 (Figure 6B).

The aboveground ABA content of ZZ39 and JZ311 seedlings increased by 1.9 and 2.2 times, respectively, after 2 days of root shearing compared with day 0. However, the ABA content decreased by 25.1% and 37.1% at 4 days compared with 2 days for ZZ39 and JZ311 seedlings, respectively (Figure 6C). After root shearing, the aboveground ZR content of ZZ39 seedlings increased by 20.0% at 4 days, and for JZ311, it increased by 17.4% at 2 days, compared with their respective levels before cutting (Figure 6D).

### 3.6. Changes in Hormone Content of Rice Seedling Roots at Different Periods

Prior to root shearing, the GA3 content in ZZ39 roots was notably higher than in the JZ311 roots, while the ZR content was significantly lower. However, the ABA and IAA content in the root systems of both varieties showed no significant difference. Two days after root shearing, the content of root GA3 and ABA in ZZ39 was significantly higher than that in JZ311, but there was no significant difference in the IAA and ZR content. After 4 days of root cutting, the levels of root GA3 and ZR were found to be highest in ZZ39 and were significantly greater than those in JZ311. However, 6 days after shearing, the levels of GA3, IAA, ABA, and ZR in JZ311 were significantly higher than those in ZZ39 (Figure 7).

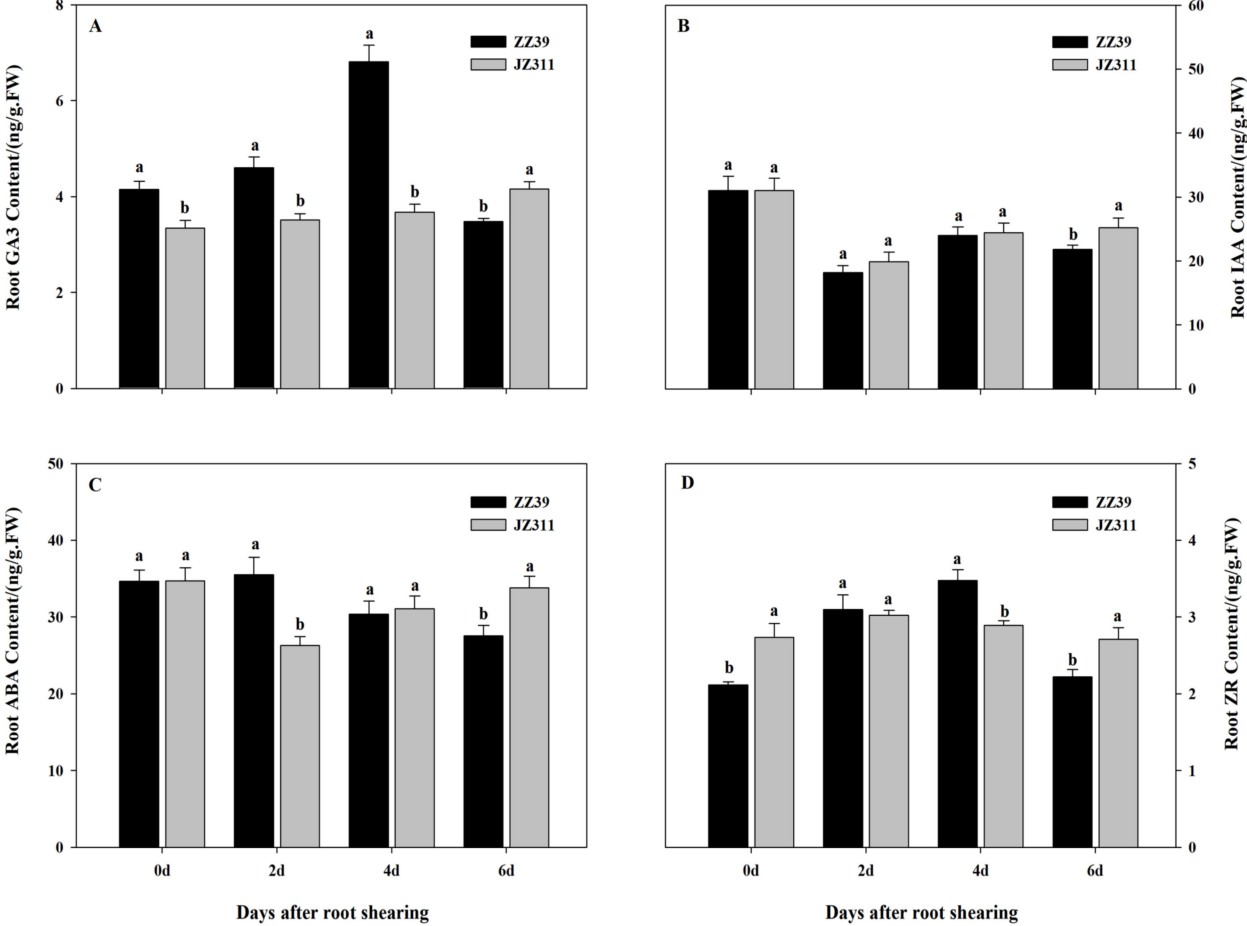

**Figure 7.** Different hormone content in the root system of rice seedlings at various periods: (**A**) the GA3 content of ZZ39 and JZ311 on days 0–6; (**B**) the IAA content of ZZ39 and JZ311 on days 0–6; (**C**) the ABA content of ZZ39 and JZ311 on days 0–6; (**D**) the ZR content of ZZ39 and JZ311 on days 0–6. The columns with the same letter are not significantly different at $p < 0.05$.

In the roots of ZZ39, the GA3 content showed a steady increase from day 0 to 4 and a significant spike 4 d after shearing, with a remarkable increase of 64.1% compared with day 0. However, there was a noticeable decline at 6 d after shearing, with a 48.9% decrease

being observed from the day 4 measurement. The content of CA3 in JZ311 roots showed a continuous increase from day 0 to 6, with a respective elevation of 5%, 9.9%, and 24.4% compared with the initial measurement (Figure 7A). After four days of ZZ39 root shearing, there was a significant increase of 32.1% in the IAA content in the new roots compared with after two days; however, six days after cutting, there was a decrease of 9.1% compared with day four. The level of IAA in the newly grown roots of JZ311 showed a steady increase from 2 to 6 days after shearing. In comparison with the levels observed at day 2, there were significant increases of 22.9% and 27% (Figure 7B).

The ABA content in ZZ39 seedling roots decreased by 12.3% and 20.5% 4 and 6 days after shearing, respectively, when compared with the content at day 0. The levels of ABA in the new roots of JZ311 seedlings increased by 18.3% and 28.4% after 4 and 6 days, respectively, compared with the levels observed 2 days after the shearing (Figure 7C). The content of ZR in the new roots of ZZ39 seedlings showed a significant increase of 46.6% and 64.3% 2 and 4 days after shearing, respectively, when compared with the content on day 0. The ZR content of new roots in JZ311 seedlings initially increased by 10.5% 2 days after shearing compared with day 0; however, it then continuously decreased by 4.4% and 10.5% compared with the content on day 2 (Figure 7D).

## 4. Discussion

The growth and development of rice is influenced by appropriate levels of N, P, and K, which have a significant impact on root length, number, and weight [29–32]. When the root system of rice is damaged, it not only impairs its normal growth and development, but also poses a threat to the aboveground part of the seedling. The impaired roots face difficulties in absorbing water and nutrients, which are crucial for the optimal development of the plant [33]. A previous study demonstrated that the aboveground N accumulation in seedlings decreased after root shearing, with a greater degree of root shearing leading to a greater decrease in N accumulation [34]. This study show that, after root shearing, the aboveground N content and P content of seedlings in both rice varieties decreased and reached their lowest level after 4 days (Figure 4A,B). This is consistent with previous studies. We also found that during the 2- to 6-day period, the length of the seedling roots, numbers of root tips, root surface area, root volume, and root dry matter weight all incrementally increased. Meanwhile, the highest growth rate was observed 4 days after shearing and hydroponics (Figures 1B and 2A–D). These results indicate that cutting off the roots of seedlings results in a temporary loss of their ability to absorb nutrients. To mitigate the damage caused by this, nutrients such as nitrogen and phosphorus present above the ground are quickly transported to the roots. This facilitates the development of new roots as soon as possible. Previous research has shown that potassium can enhance plant stress resistance [35,36]. In our study, the potassium content above the ground of ZZ39 and JZ311 increased at different rates after root shearing, with ZZ39 showing a greater increase at 4 days and JZ311 showing a greater increase at 2 days (Figure 4C), suggesting that JZ311 has faster responsiveness and self-healing abilities in response to root damage. Throughout all periods, ZZ39 had a significantly higher aboveground potassium content than JZ311, indicating that ZZ39 was more resilient to stress than JZ311.

During rooting, the aboveground soluble sugars in seedlings are decomposed through oxidation, providing energy and structural material for root growth and development; consequently, there is a significant decrease in their content [37]. Soluble sugars and starch are the primary carbon sources in rice that have a significant impact on metabolic activities. These compounds serve as energy sources and can enhance plant resilience during periods of stress [38–41]. The soluble sugars can be transported to cells or organs in need through cell interstitial spaces or cell membranes, providing them with the necessary energy [42]. This study found that the levels of soluble sugar and starch in the aboveground parts of ZZ39 and JZ311 were markedly reduced within two days of root shearing (Figure 3A,B). Notably, the seedling roots regrew during this period, suggesting that the carbohydrates present in the aboveground parts were quickly transferred to the seedling roots to provide energy for their early root development [43]. In this study, the most significant period

of increase in new root parameters for both rice varieties occurred on the fourth day of post-cutting hydroponics. At the same time, we found that the soluble sugar content and starch content in the aboveground parts of the seedlings reached their peak (Figure 3A,B), indicating that the rapid growth of new roots promotes photosynthesis, leading to increased carbohydrate production in the aboveground parts [44]. Notably, the soluble sugar and starch content of both varieties of seedlings significantly decreased the sixth day after root shearing due to the increase in new roots (Figure 3A,B). We speculate that this is likely due to a decrease in chlorophyll content and subsequent decrease in photosynthetic products, as well as an increase in the metabolic activities and energy consumption of the plants after the recovery of the injured root system. Previous studies have shown that when plants are under stress, they activate a range of responses to protect themselves and enhance their ability to adjust to environment [45–47]. This study found that ZZ39 exhibited a notable rise in chlorophyll-a levels on the second day after root shearing, which was followed by a rapid decline (Figure 5A), indicating that it is a mechanism employed by the plant to counteract the stress caused by root injury, whereby photosynthesis is boosted for a brief period to aid in recovery.

The regulation of new root growth in rice is influenced by endogenous hormones, which also significantly contribute to seedling growth and development, environmental adaptation, and metabolic activities [48,49]. Recent research has demonstrated that auxin plays a vital role in the transportation and distribution of nutrients and energy materials [50]. This is achieved through the regulation of various cellular processes such as cell wall relaxation, cell membrane permeability, and vascular bundle development and differentiation, as well as photosynthesis and respiration in leaves and fruits [51]. We found that the aboveground IAA content of JZ311 seedlings experienced a sharp increase and peaked two days after root shearing (Figure 6B). However, the soluble sugar content and starch content significantly decreased during this period and were the lowest among all treatment periods (Figure 3A,B). Based on the information mentioned above, we speculate that this phenomenon is likely due to the urgent need for nutrients and energy materials in the root system to repair damaged organs. The increase in auxin content accelerates the rate of transport of these materials from the ground to the roots, effectively promoting the early development of new roots. Furthermore, the rice root system has a dual purpose: it absorbs nutrients and water while also synthesizing various hormones, organic acids, and amino acids [7,52]. The production of GA3 and ZR is particularly noteworthy, as GA3 and ZR significantly promote seedling roots' growth and development [53]. Our study revealed significant growth in the root length, number, and dry matter weight after ZZ39 root shearing, specifically after four days (Figures 1B and 2A,B). This increase was attributed to the improved self-synthesis ability of GA3 and ZR after new root production. Following root damage, previous studies have discovered that TAA family genes facilitate the continuous production of the intermediate product IPA [54,55]. Subsequently, YUC family genes respond to wound signals, leading to the induction of IAA production, which ultimately promotes the germination of new roots [56,57]. Moreover, plants must develop a quick self-test regeneration mechanism due to their immobility. This mechanism can reactivate developmental signaling pathways at the injury site, which facilitates the repair and regeneration of necrotic or lost cells [58–60]. Plant trauma serves as a signal that triggers the production of auxin in the chloroplasts of leaf explants. The auxin is then transported via polar transport to receptor cells near the wound, where it accumulates to high levels and activates regeneration or repair mechanisms [22].

According to Matosevich et al. [61], the rapid accumulation of auxin near the site of injury is the cause of tissue or organ regeneration in plants. Meanwhile, we found that there was a noteworthy rise in IAA content in the new roots of ZZ39 and JZ311 within 2–4 days after root shearing (Figure 7B). This increase led to a significant improvement in all parameters of the root system (Figure 2A–D). These results suggest that injured seedling roots can rapidly accumulate IAA for a short period, leading to the production and rapid growth of new roots.

## 5. Conclusions

In conclusion, our results indicate that the ability of rice seedlings to regenerate roots significantly vary among different varieties. While ZZ39 exhibited stronger root-related traits than JZ311 before root shearing, JZ311 showed stronger root-generating ability after root shearing. The period between 2 and 4 days after root cutting was identified as the "outbreak" period for new roots in both varieties, with a sharp increase in all parameters of new roots. The physiological characteristics of seedlings after root shearing indicate that nutrients such as aboveground nitrogen, soluble sugars, and starch are transported to the injured roots to provide energy and nourishment for new root production. The increase in hormone levels aboveground in cut-root seedlings facilitates the transfer of nutrients and carbohydrates from the stems and leaves to the roots. Additionally, damaged parts of the plant roots accumulate higher concentrations of IAA in a short period of time. The process of dual regulation, both above and below the ground, helps to restore and activate damaged cells within the root system. As a result, this process ultimately leads to the emergence of new roots. This study utilized hydroponics, which presents certain limitations due to its significant differences from the natural field environment. As a signaling molecule, reactive oxygen species (ROS) plays an important role in regulating the response of plants to abiotic stresses; thus, the effect of ROS on new root production should be further investigated in the future.

**Author Contributions:** J.X., H.C. and Y.Z. (Yunbo Zhang) conceived the project and designed the experiments; Y.G. and Y.Z. (Yan Zhu) performed the experiments; Y.G., J.X., Y.Z. (Yikai Zhang), Y.W., Y.Z. (Yunbo Zhang) and Z.W. analyzed the data; Y.Z. (Yuping Zhang), H.C. and J.X. contributed reagents/materials/analysis tools; Y.G. and J.X. drew the figures and wrote the manuscript. H.C. and Y.Z. (Yunbo Zhang) revising the manuscript. All authors have read and agreed to the published version of the manuscript.

**Funding:** This research was funded by the National Key Research and Development Plan of China under "Research and Demonstration of Integrative Approaches to Synergistically Improve Yield, Quality and Efficiency in Rice Production in Southern China" (2022YFD2300700); the Research and Development Project of Zhejiang Province (2022C02048); the Agricultural Sciences and Technologies Innovation Program of the Chinese Academy of Agricultural Sciences (CAAS) for the Rice Intelligence and High Efficiency Cultivation Technology Group; and the Special Funds for the Construction of Modern Agricultural Technology System (CARS-01-23).

**Data Availability Statement:** Date sharing not applicable.

**Conflicts of Interest:** The authors declare no conflict of interest.

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
