# Peer review of "Physiological Characteristics of Root Regeneration in Rice Seedlings"

_agronomy, doi:10.3390/agronomy13071772_

Round 1

Reviewer 1 Report

The article “Physiological characteristics of root regeneration in rice seedlings” identified key physiological characteristics that affect new root generation, providing a scientific basis for identifying strong root regeneration varieties and developing cultivation measures to promote new root growth in rice. However, there are some shortcomings which must be revised.

Add brief methods and main or specific findings of the study.

What is ZZ39 and JZ11. Explain for the readers at first use.

Line 51-52 lack references and should be cited with recent study https://doi.org/10.1016/j.xplc.2023.100597, https://doi.org/10.1007/s10725-021-00785-7

The introduction is too long and have no coherence.

One of the established physiological characteristics of root regeneration in rice seedlings is the role of auxin in initiating root regeneration. Auxin is a plant hormone that plays a critical role in root development and regeneration. After root damage or stress, auxin levels increase in the plant, which triggers the formation of new roots. Also discuss the role of such hormones briefly in the paragraph of regeneration.

Line 85-88, which numerous studies, numerous studies should be cited. The following studies can be helpful doi: 10.1038/s43016-021-00300-1, https://doi.org/10.1007/s10534-022-00417-1

All abbreviations must be cited at first use.

Involvement of reactive oxygen species (ROS) in root regeneration. ROS are molecules that are produced in response to stress or damage and can have both positive and negative effects on plant growth. In rice seedlings, ROS can promote root regeneration by signaling the expression of genes involved in cell proliferation and differentiation. But the study did not performed any experiment related to ROS in roots.

In discussion every results should be discussed by comparing with relevant studies.

Discussion look like introduction. Please discuss results.

Add future perspective and study gap in the conclusion. One paragraph conclusion is enough.

Lengthy sentences should be revised. write short sentences to convey clear message of the sentence

Author Response

Dear Editor:

We are very grateful to the reviewers’ comments for the manuscript. According to your suggestions, we have revised the manuscript. Below are the responses to each question:

Q1: Add brief methods and main or specific findings of the study.

A: We have added brief methods and main or specific findings of the study.

Q2: What is ZZ39 and JZ11. Explain for the readers at first use.

A: We have explained to readers in the article.

Q3: Line 51-52 lack references and should be cited with recent study

A: Thank you for pointing this out! We have introduced the recent relevant references into the article. (Line 52-53)

Q4: The introduction is too long and have no coherence.

A: Thank you! After revising the introduction, we eliminated irrelevant content and improved the coherence and conciseness of the article.

Q5: One of the established physiological characteristics of root regeneration in rice seedlings is the role of auxin in initiating root regeneration. Auxin is a plant hormone that plays a critical role in root development and regeneration. After root damage or stress, auxin levels increase in the plant, which triggers the formation of new roots. Also discuss the role of such hormones briefly in the paragraph of regeneration.

A:We have discussed the role of auxin in the paragraph of regeneration briefly. (Line71-74)

Q6: Line 85-88, which numerous studies, numerous studies should be cited.

A:We have introduced relevant literature in this section.(Line 77)

Q7: All abbreviations must be cited at first use.

A:All abbreviations have been cited at first use.

Q8: Involvement of reactive oxygen species (ROS) in root regeneration. ROS are molecules that are produced in response to stress or damage and can have both positive and negative effects on plant growth. In rice seedlings, ROS can promote root regeneration by signaling the expression of genes involved in cell proliferation and differentiation. But the study did not performed any experiment related to ROS in roots.

A:Thank you for pointing this out! The aim of our experiments is to investigate the impact of nutrients, energy, and hormones on the production of new roots in seedlings. However, we did not explore the correlation between ROS and the emergence of new roots in this particular experiment. This is an area we plan to address in future experiments.

Q9: In discussion every results should be discussed by comparing with relevant studies. Discussion look like introduction. Please discuss results.

A:In the discussion section, we have made revisions based on your comments. We have discussed each result by comparing relevant studies.

Q10: Add future perspective and study gap in the conclusion. One paragraph conclusion is enough.

A:We have added future perspective and study gap in the conclusion. “This study utilized hydroponics as a method, which presents certain limitations due to its significant differences from the natural field environment. As a signaling molecule, reactive oxygen species (ROS) plays an important role in regulating the response of plants to abiotic stresses, so the effect of ROS on new root production should be further investigated in the future. ”

We hope that we have revised the manuscript to the reviewer’s request. Should you need any further information and clarification, please do not hesitate to contact us. Many thanks for your time and consideration!

Best wishes,

Jing Xiang

State Key Laboratory of Rice Biology and Breeding, China National Rice Research Institute, Hangzhou, Zhejiang 310006, China, E-mail: xiangjing_823@163.com, Tel: +008618958071661

Reviewer 2 Report

Congratulations on completing the study. However, the materials and methods section requires thorough revision. Too much missing information and confusing sentences. 

Line 104-105: Times symbol, 3mm change to 3 mm

Line 104: Rephrase "consistent germination rice seeds"

Line 105: 48 seeds for each time treatment? Please clarify or rephrase the sentence. 

Line 111: is the 0d used as control? 

Line 123-124: very confusing sentence. How author replicate "them" 

Line 133-135: Please elaborate more on chlorophyll content determination. Wavelength used for chlorophyll a and b and also the carotenoid?

Line 139: 2.4 Statistical analysis. The authors should improve this part

Line 141: state clearly what kind of analysis was carried out using SPSS

It is clear based on the figures that different type of analysis was carried out. Authors also used alphabets to indicate significant differences at p<0.05 but what type of post-hoc test was done? Too much information was missing in the analysis section

Line 515: Conclusion not really conclusive. Any take home message for readers?

Minor editing is required.

Author Response

Dear Editor:

We are very grateful to the reviewers’ comments for the manuscript. According to your suggestions, we have revised the manuscript. Below are the responses to each question:

Q1: Times symbol, 3mm change to 3 mm

A: Thank you! We have made revisions (Line 96)

Q2: Rephrase“consistent germination rice seeds”

A: We have revisedconsistent germination rice seedstoRice seeds with consistent germination.(Line 95)

Q3: 48 seeds for each time treatment? Please clarify or rephrase the sentence.

A: We have rephrased this sentence in the experimental design in order for the reader to better understand the experiment. ( Line 96-97)

Q4: is the 0d used as control?

A: Yes, the 0d is used as control.

Q5: very confusing sentence. How author replicate ”them“

A: In each period, we selected thirty seedlings of each variety with uniform growth and each ten seedlings were used as one replicate. (Line 114-115)

Q6: Please elaborate more on chlorophyll content determination. Wavelength used for chlorophyll a and b and also the carotenoid?

A: In this study, we extracted Chlorophyll and carotenoids from 0.1 g fresh rice seedlings using 96% ethanol. The absorbance of the extract was measured at 665, 649, and 470 nm using a spectrophotometer, following the method described by Lichtenthaler and Wellburn. (Line 124-127)

Q7: Statistical analysis. The authors should improve this part

A: Thank you for pointing this out! We have further refined our statistical analysis. ( Line 131-135)

Q8: state clearly what kind of analysis was carried out using SPSS

A: SPSS 23.0 statistical software was used to perform three repeated one-way analyses of variance (ANOVA), Duncan’s multiple comparisons.

Q9: It is clear based on the figures that different type of analysis was carried out. Authors also used alphabets to indicate significant differences at p<0.05 but what type of post-hoc test was done? Too much information was missing in the analysis section.

A: Thank you for pointing this out! We used post-hoc tests done by Duncan in SPSS.

Q10: Conclusion not really conclusive.Any take home message for readers?

A: In this study, we conducted an experiment to screen rice varieties with robust root regeneration. Our results revealed key physiological factors that influence the emergence of new roots. These results offer a scientific and theoretical foundation for addressing the issue of slow green turning in machine-planted rice seedlings with damaged roots.

We hope that we have revised the manuscript to the reviewer’s request. Should you need any further information and clarification, please do not hesitate to contact us. Many thanks for your time and consideration!

Best wishes,

Jing Xiang

State Key Laboratory of Rice Biology and Breeding, China National Rice Research Institute, Hangzhou, Zhejiang 310006, China, E-mail: xiangjing_823@163.com, Tel: +008618958071661

Reviewer 3 Report

It is interesting and useful that the authors have investigated the dynamic changes in new root production and growth of seedlings of two varieties after root cutting, as well as key physiological factors. In general, the MS was written sound. Hence, it is recommended to be accepted after some revisions.

1.       Add some basic information of two rice varieties, and indicate the reasons why they were selected for the experiment.

2.       Present used one-way or two-way ANOVA for the variance of the data for each treatment.

Minor editing of English language required.

Author Response

Dear Editor:

We are very grateful to the reviewers’ comments for the manuscript. According to your suggestions, we have revised the manuscript. Below are the responses to each question:

Q1: Add some basic information of two rice varieties, and indicate the reasons why they were selected for the experiment.

A: Thank you! We have added to the material methodology about the reasons why two rice varieties were selected as test material(Line 89-90).

Q2: Present used one-way or two-way ANOVA for the variance of the date for each treatment.

A: We used a one-way ANOVA for the variance of the data for each treatment period in present.

We hope that we have revised the manuscript to the reviewer’s request. Should you need any further information and clarification, please do not hesitate to contact us. Many thanks for your time and consideration!

Best wishes,

Jing Xiang

State Key Laboratory of Rice Biology and Breeding, China National Rice Research Institute, Hangzhou, Zhejiang 310006, China, E-mail: xiangjing_823@163.com, Tel: +008618958071661